# Plasma lipidome profiling of newborns with antenatal exposure to Zika virus

**Nieli Rodrigues da Costa Faria**[1☺], **Adriano Britto Chaves-Filho**[2☺], **Luiz Carlos Junior Alcantara**[1], **Isadora Cristina de Siqueira**[3], **Juan Ignacio Calcagno**[4], **Sayuri Miyamoto**[2], **Ana Maria Bispo de Filippis**[1], **Marcos Yukio Yoshinaga**[2☺]*

1 Flavivirus Laboratory, Oswaldo Cruz Institute (FIOCRUZ), Rio de Janeiro, Brazil, 2 Department of Biochemistry, Institute of Chemistry, University of São Paulo, São Paulo, Brazil, 3 Gonçalo Moniz Institute, FIOCRUZ, Bahia, Brazil, 4 Maternidade Prof. José Maria de Magalhães Netto, State Health Secretary (Salvador), Bahia, Brazil

☺ These authors contributed equally to this work.
* marcosyukio@gmail.com

**Data Availability Statement:** All relevant data are within the manuscript and its Supporting Information files.

## Abstract

The 2015–2016 Zika virus (ZIKV) outbreak in Brazil was remarkably linked to the incidence of microcephaly and other deleterious clinical manifestations, including eye abnormalities, in newborns. It is known that ZIKV targets the placenta, triggering an inflammatory profile that may cause placental insufficiency. Transplacental lipid transport is delicately regulated during pregnancy and deficiency on the delivery of lipids such as arachidonic and docosahexaenoic acids may lead to deficits in both brain and retina during fetal development. Here, plasma lipidome profiles of ZIKV exposed microcephalic and normocephalic newborns were compared to non-infected controls. Our results reveal major alterations in circulating lipids from both ZIKV exposed newborns with and without microcephaly relative to controls. In newborns with microcephaly, the plasma concentrations of hydroxyoctadecadienoic acid (HODE), primarily as 13-HODE isomer, derived from linoleic acid were higher as compared to normocephalic ZIKV exposed newborns and controls. Total HODE concentrations were also positively associated with levels of other oxidized lipids and several circulating free fatty acids in newborns, indicating a possible plasma lipidome signature of microcephaly. Moreover, higher concentrations of lysophosphatidylcholine in ZIKV exposed normocephalic newborns relative to controls suggest a potential disruption of polyunsaturated fatty acids transport across the blood-brain barrier of fetuses. The latter data is particularly important given the neurocognitive and neurodevelopmental abnormalities observed in follow-up studies involving children with antenatal ZIKV exposure, but normocephalic at birth. Taken together, our data reveal that plasma lipidome alterations associated with antenatal exposure to ZIKV could contribute to identification and monitoring of the wide spectrum of clinical phenotypes at birth and further, during childhood.

## Author summary

Antenatal exposure to Zika virus (ZIKV) is linked to a wide range of clinical presentations at birth, from asymptomatic cases to microcephaly, and other neurocognitive and

**Funding:** This research was funded by the Coordenação de Vigilância em Saúde e Laboratórios de Referência, Brazilian Ministry of Health (CVSRL-Fiocruz to AMBF), Fundação Carlos Chagas Filho de Amparo à Pesquisa do Estado do Rio de Janeiro (FAPERJ, grant number E-26/2002.930/2016 to AMBF), Horizon 2020 through ZikaPlan and ZikAction (grant agreement numbers 734584 and 734857), the International Development Research Centre Canada (IDRC, 108411-001 to AMBF), the National Council for Scientific and Technological Development (CNPq, 440685/2016-8 and 443875/2018 to ICS), and the São Paulo Research Foundation (FAPESP, CEPID-Redoxoma 2013/07937–8 to SM). Postdoc fellowships were received from the Coordination for the Improvement of Higher Education Personnel (MYY), IDRC (NRCF) and FAPESP (ABC-F). The funders had no role in study design, data collection and analysis, decision to publish, or preparation of the manuscript.

**Competing interests:** The authors have declared that no competing interests exist.

neurodevelopmental abnormalities manifested in the early childhood. Stratification of these clinical phenotypes in newborns with suspected antenatal ZIKV exposure is challenging, but critical to improve early assessment of rehabilitative interventions. In this study, plasma lipidome profiling of 274 lipid species was performed in both normocephalic and microcephalic newborns with antenatal ZIKV exposure and compared to non-infected controls. Multiple lipid species were independent predictors of antenatal ZIKV exposure. More specifically, microcephaly was strongly associated with an oxidized free fatty acid and ZIKV exposed normocephalic newborns exhibited higher plasma concentrations of lysophosphatidylcholine relative to controls. These findings emphasize the need for studies focused on the role of individual lipids in neuropathogenesis of ZIKV and raise the potential of plasma lipidome profiling for early diagnosis of newborns with suspected antenatal ZIKV exposure. To validate the predictive ability of this approach, prospective studies with a larger cohort of newborns are now required.

## Introduction

Exposure of newborns to Zika virus (ZIKV) during pregnancy has been linked to congenital ZIKV syndrome (CZS), resulting in severe neurodevelopmental abnormalities in infants, most prominently microcephaly, with other associated clinical presentations such as seizures, hearing and visual abnormalities, dysphagia and fetal death [1–6]. A case series study with 182 symptomatic ZIKV-infected pregnant women in Brazil revealed an expressive 42% of fetuses presenting abnormal clinical or brain imaging outcomes, regardless of the trimester of infection [1]. In a larger study (>2,000 pregnancies) including all the United States territories, an overall estimate of 5% of fetuses or infants with birth defects was also reported independently of the trimester of infection [7]. Antenatal ZIKV exposure may not manifest as CZS in infants, as a broad spectrum of clinical presentations from asymptomatic to microcephaly may occur. Longitudinal cohort studies with CZS-affected children followed for 8 to 24 months after birth have shown that the majority of participants had major abnormalities related to, among others, irritability, seizure disorders and severe motor impairment [8,9]. Normocephalic ZIKV exposed newborns may also develop significant abnormalities 1 to 3.5 years after birth as revealed by brain imaging, neurodevelopmental, neurocognitive and ophthalmological evaluations [2,10–13].

The molecular mechanisms by which ZIKV harms the developing brain remain unknown, but it is well established that ZIKV infection impairs multiplication and migration of the human cortical neural progenitor cells [14–16]. Transmission of ZIKV to fetuses must occur via placenta [17,18], and there exist robust clinical and experimental evidence for ZIKV targeting placental cells, including trophoblasts, Hofbauer macrophages and fetal endothelial cells [19–25]. The maternal-fetal interface is of critical importance for embryonic development given the transfer of energy, signals and nutrients (e.g. glucose, amino acids and lipids) from the mother's bloodstream [26]. Alterations in placental inflammatory profiles due to ZIKV infection have been reported in cell culture studies [19,24,27] and elevated concentrations of inflammatory markers have been detected in cord blood plasma of newborns exposed to ZIKV [28]. ZIKV infection may also trigger placental metabolic reprogramming, resulting in *de novo* lipogenesis with a remarkable accumulation of cytosolic lipid droplets [29].

Changes in placental lipid metabolism are likely to impact the embryonic/fetal development, especially the brain and eye that are highly depend on the transfer of polyunsaturated fatty acids, such as docosahexaenoic acid (DHA), across the placenta. Any disruption in the

uptake of these polyunsaturated fatty acids may lead to several brain and eye damage in infants [30–32]. For instance, inactivating mutations of the human major facilitator superfamily domain-containing protein 2 (Mfsd2a)–a major transporter of DHA to the brain [33]–were reported to cause lethal to mild microcephaly in humans [34,35]. Interestingly, ZIKV was recently reported to disrupt Mfsd2a both in human brain endothelial cell cultures and neonatal mouse brain, causing fetal growth restriction and microcephaly in the latter [36].

This study sought to provide insights into the detection of clinical phenotypes derived from antenatal ZIKV exposure. For this purpose, plasma lipidome profiles of ZIKV exposed newborns with microcephaly and normocephaly were compared to those of non-infected controls.

## Methods

### Ethics statement

The study was approved by the Institutional Review Board of the Oswaldo Cruz Foundation (FIOCRUZ), Rio de Janeiro, Brazil (CAAE: 90249218.6.1001.5248) and by Gonçalo Moniz Institute, FIOCRUZ local Ethics Committee (CAAE: 51889315.7.0000.0040). The legal guardians of all newborns enrolled in this study provided written informed consent.

### Subjects' recruitment, sample collection and ZIKV diagnosis

Participants were recruited from a previous neonatal surveillance for congenital Zika infection from January to December 2016 at the José Maria Magalhães Netto public maternity hospital located in Salvador [37], one of the most relevant Brazilian cities during the microcephaly outbreak.

Thirty participants were enrolled in this study: 10 control/healthy newborns without ZIVK infection (G1), 9 normocephalic newborns exposed to ZIKV (G2) and 11 newborns with ZIKV-induced microcephaly (G3). Clinical and epidemiological data of newborns were obtained through interviews with mothers and review of medical records. Data storage and management was performed using the REDCap 6.18.1 (Vanderbilt University, Nashville, TN). All enrolled newborns were classified according to the International Fetal and Newborn Growth Consortium for the 21st Century (INTERGROWTH-21st) charts, taking into account the newborn's gender, gestational age and head circumference at birth [38]. Microcephaly was defined as head circumference measuring less than two standard deviations below the average, while severe microcephaly was considered if head circumference measurements were less than three standard deviations below the average. Newborns were considered normocephalic, if their head circumference measurements were within two standard deviations. These data are reported in S1 Table.

All samples were collected at birth from the umbilical cord vein. Blood samples were obtained in ethylenediaminetetraacetic acid (EDTA) tubes and plasma was obtained by centrifugation and stored at −80˚C. Serological analyses and molecular diagnosis for ZIKV were performed according to previously described methods [37,39,40]. Congenital ZIKV infection was defined as newborns whose serological testing (anti-ZIKV immunoglobulin M) or a qualitative reverse-transcription polymerase chain reaction assay for ZIKV was positive. Healthy controls had negative serological and molecular results for ZIKV (see S1 Table). All newborns' samples were negative when tested for syphilis, HIV, toxoplasma (IgM) and cytomegalovirus (IgM).

### Plasma lipidome analysis

An aliquot of 20 μL of plasma or 20 uL of water (extraction blanks) were spiked with 50 μL of an internal standard mixture (Table 1) and total lipid extraction was performed as previously

described [41] (see details in S1 Methods). Simultaneously, a pooled aliquot of all samples was extracted and used as quality controls for reproducibility test.

Lipid analysis was performed by an untargeted lipidomics approach [42,43] using an ultra high-performance liquid chromatography (UHPLC Nexera, Shimadzu, Kyoto, Japan), electro-spray ionization tandem time-of-flight mass spectrometry (ESI-Q-TOFMS, Triple TOF 6600, Sciex, Concord, US). Chromatographic and mass spectrometry conditions are provided in Supplementary Methods. In brief, the MS operated in both positive and negative ionization modes, with a scanning range of 200–2000 Da, and samples were randomly analyzed (1μL injection volume) with a control and a blank sample analyzed within each batch of 5 experimental samples. At least two samples per group were used for MS/MS identification, with inspection of 400 and 300 ions in negative and positive ionization modes, respectively. Lipid molecular species were manually annotated exclusively based on their exact masses coupled to specific MS/MS fragments and/or neutral losses obtained by Information Dependent Acquisition (IDA) as outlined elsewhere [44]. The exceptions were free fatty acids and free cholesterol from which MS/MS data is poorly observable and therefore were identified based on exact masses and retention time.

Quantification of lipid molecular species was performed by comparison of chromatographic peaks of precursor ions (MS1) to those of the corresponding internal standard (Table 1), using 5 mDa as limit for attribution. The integral lipidomics dataset is provided in S2 Table as area ratio. Results were expressed in mg/dL of plasma to facilitate comparison with the literature or ng/μL otherwise, while fatty acid composition of total lipids and classes were calculated in molar concentrations. Data are presented as average ± standard error of the

**Table 1. Lipid classes, number of lipid species and internal standards used for semi-quantification by untargeted lipidomics.**

| Lipid classes | # species | Internal standards |
|---|---|---|
| Monohexosyl ceramide (1H-Cer) | 4 | Cer (d18:1/17:0) |
| Dihexosyl ceramide (2H-Cer) | 1 | Cer (d18:1/17:0) |
| Trihexosyl ceramide (3H-Cer) | 1 | Cer (d18:1/17:0) |
| Ceramides (Cer) | 18 | Cer (d18:1/17:0) |
| Sphingomyelin (SM) | 20 | SM (d18:1/17:0) |
| Lysophosphatidylcholine (LPC) | 8 | LPC (17:0) |
| Alkanyl-phosphatidylcholine (oPC) | 6 | PC (17:0/17:0) |
| Phosphatidylcholine (PC) | 30 | PC (17:0/17:0) |
| Alkenyl-phosphatidylcholine (pPC) | 5 | PC (17:0/17:0) |
| Lysophosphatidylethanolamine (LPE) | 2 | LPE (17:1) |
| Alkanyl-phosphatidylethanolamine (oPE) | 1 | PE (17:0/17:0) |
| Phosphatidylethanolamine (PE) | 9 | PE (17:0/17:0) |
| Alkenyl-phosphatidylethanolamine (pPE) | 10 | PE (17:0/17:0) |
| Phosphatidylinositol (PI) | 8 | PC (17:0/17:0)* |
| Free fatty acids (FFA) | 21 | FFA (13:0) |
| Oxidized fatty acids (Oxy-FA) | 3 | FFA (13:0) |
| Acylcarnitines (AC) | 13 | LPC (17:0)** |
| Free cholesterol (FC) | 1 | CE (22:0) |
| Cholesteryl esters (CE) | 14 | CE (22:0) |
| Triglycerides (TG) | 99 | TG (17:0/17:0/17:0) |
| Total: 20 classes | Total: 274 | All standards at 10 μg/mL |

*a response factor of 0.65 was applied for PI concentrations based on an external calibration (see S1 Methods).

**AC were quantified based on LPC (17:0) as reference.

mean. Note that internal standards for some lipid classes (e.g. mono- and di-hexosyl ceramides, coenzyme Q10 and acylcarnitines; Table 1) were unavailable in the laboratory and thus their concentrations are not comparable to other compounds, but largely comparable among samples. Values for each lipid class were calculated as the sum of the individual lipid species.

### Statistical analysis

Statistical analyses of data obtained from untargeted lipidomics were performed with Metaboanalyst (www.metaboanalyst.ca; according to [45]). Prior to statistical analyses data, the coefficient of variance (CV) was calculated for each quantified lipid in quality control samples, and lipid species displaying CV values above 20% were excluded from further analyses. The data were log-transformed, and pairwise comparisons performed by multivariate (orthogonal partial least square discriminant analysis) and univariate (unpaired t-test) analyses. After checking for data distribution, a Spearman's rank correlation analysis was applied.

## Results

In this study, a total of 30 participants belonging to three groups were enrolled: 10 control/healthy newborns without antenatal ZIVK exposure, 9 normocephalic newborns exposed to ZIKV and 11 newborns with ZIKV-induced microcephaly (denominated G1, G2 and G3, respectively). The characteristics of the participants are delineated in S1 Table. The present study identified and quantified 274 individual species of lipids in plasma that were distributed into 20 classes/subclasses according to Table 1. In terms of concentrations, major lipid pools were represented by cholesteryl esters (CE), followed by triglycerides (TG), phosphatidylcholine, sphingomyelin and free fatty acids (FFA), and displayed a considerable variation in lipid concentrations within groups (S1A Fig). Concentrations of both TG and CE were comparable to the values reported in the literature [46], with significantly reduced amounts of both CE and particularly TG in newborns relative to mothers' plasma in normal pregnancy [47].

Pairwise comparisons were performed with the 274 lipids by multivariate and univariate analyses. Prior to these statistical routines, we excluded 13 lipids that displayed coefficient of variation higher than 20% in quality control samples. Orthogonal partial least square discriminant analysis provided a clear separation of groups based on the composition of plasma lipidome (Fig 1). As a common trend in the ZIKV-infected groups G2 and G3, we found lower concentrations of several CE and glycerophospholipids species (mainly phosphatidyl-ethanolamine and -inositol) and higher concentrations of FFA relative to the non-infected group G1 (Fig 1A and 1B). The ZIKV exposed groups also displayed a trend for elevated concentrations of TG species linked to docosahexaenoic (22:6) and arachidonic (20:4) acids relative to G1, whereas G1 was enriched in TG species linked to linoleic acid (18:2) relative to G2 and G3 (loadings plot in Fig 1A and 1B). The latter differences, however, were not confirmed by univariate analysis of specific fatty acids esterified to total lipids or specific classes such as TG and FFA (S1B and S1C Fig). A particular feature of G2 relative to G1 and G3 was the elevated concentrations of lysophosphatidylcholine (LPC) species (Fig 1A and 1C), while G3 displayed higher concentrations of both FFA and oxidized free fatty acids (Oxy-FA) species relative to the other groups (Fig 1B and 1C).

Further comparisons of newborns by unpaired t-test corroborated results obtained by multivariate analysis. Remarkable plasma lipidome alterations were noticed comparing the control group G1 to the ZIKV exposed groups (Fig 2). For instance, concentrations of LPC (16:1, 20:3 and 20:4) and a single TG were significantly elevated in G2 relative to G1. In contrast, free cholesterol and 13 TG (10 of them linked to at least one chain of linoleic acid) displayed lower concentrations in G2 relative to G1. Several saturated FFA (14:0, 15:0, 17:0 and 18:0) as well as

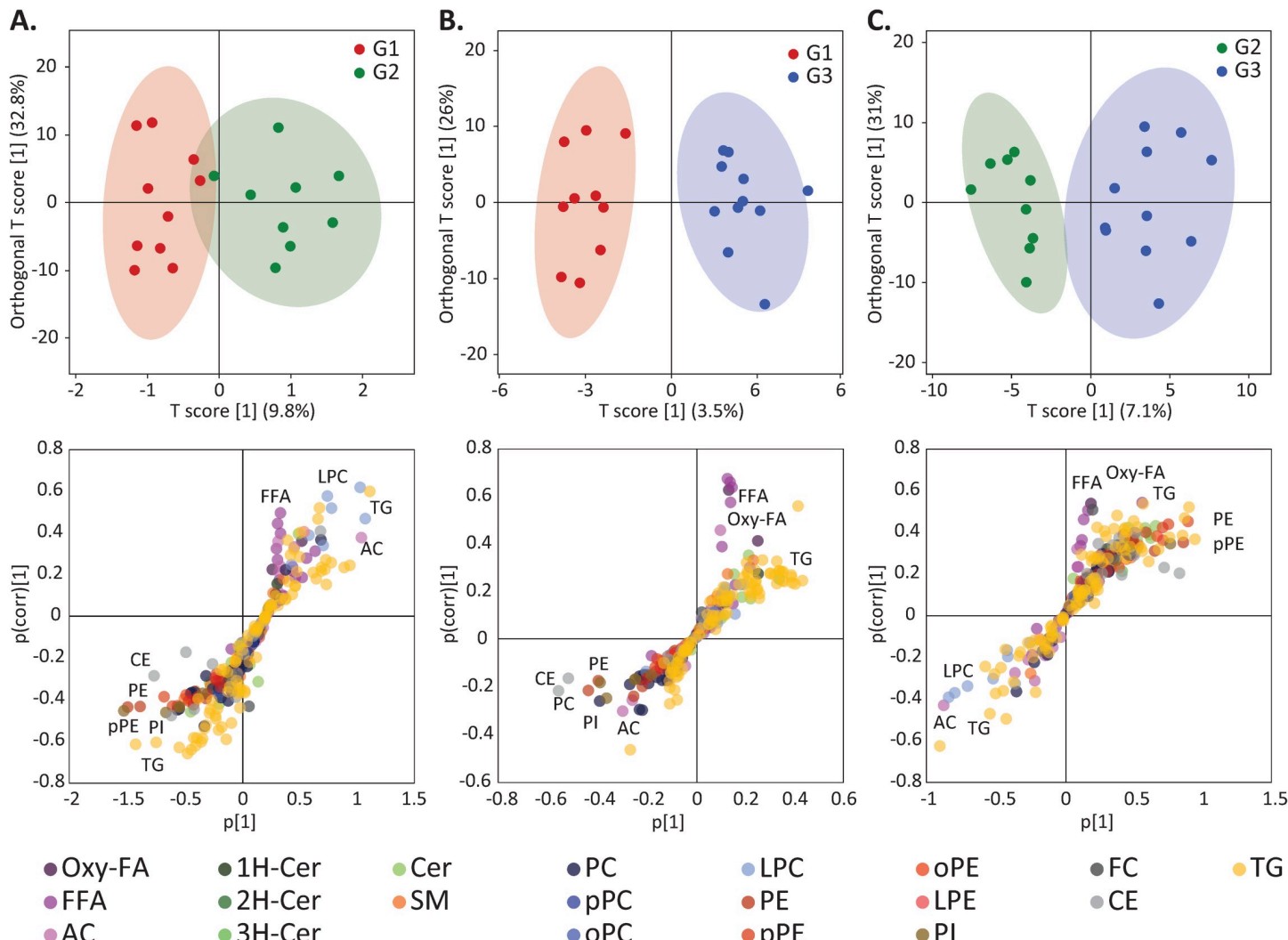

**Fig 1. Multivariate analysis by orthogonal partial least square discriminant analysis (oPLSDA) revealing discriminant features in umbilical cord plasma lipidome among the groups of newborns in pairwise comparisons.** The groups were composed of non-infected controls (G1), ZIKV-infected without and with microcephaly (G2 and G3, respectively). The upper and lower panels display, respectively, score and loadings plots of the oPLSDA for G1 versus G2 (A), G1 versus G3 (B) and G2 versus G3 (C). In lower panels, selected lipid classes leading to contrasting plasma lipidome composition among groups of newborns are highlighted. Abbreviations for lipid classes are depicted in Table 1.

acylcarnitine (AC 8:0), hydroxyoctadecadienoic acid (HODE) and TG (18:0/18:0/18:1) displayed higher concentrations in G3 relative to G1. Conversely, a single TG (16:1/18:1/18:2) was found in reduced concentrations in G3 as compared to G1.

Among the ZIKV exposed groups, higher concentrations of free cholesterol, FFA (20:1), HODE and 3 TG were found in G3 relative to G2 (Figs 2 and S2). The normocephalic ZIKV exposed group G2 showed higher concentrations of TG (18:2/20:4/20:4) and lower concentrations of free cholesterol and 3 TG (all linked to at least one linoleic acid chain) relative to the other groups. Importantly, elevated concentrations of HODE were observed in G3 relative to the other groups, suggesting this oxidized lipid derived from linoleic acid as a potential metabolic signature of ZIKV-induced microcephaly.

Because our untargeted lipidomic method was not designed to precisely estimate the abundance and composition of Oxy-FA, and given their importance in this study, we conducted a

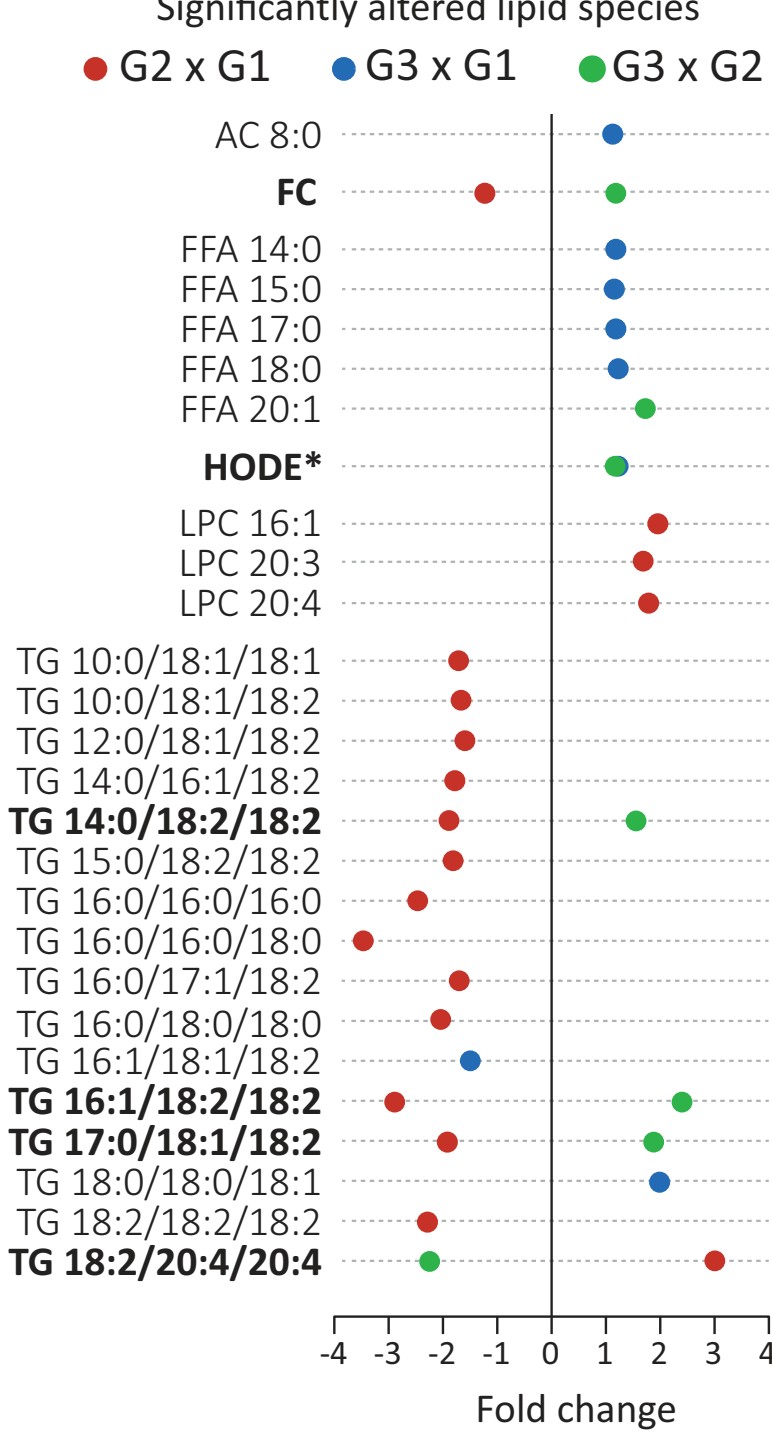

**Fig 2. Major alterations in plasma lipidome of newborns caused by ZIKV infection.** Fold-change of significantly altered lipids in pairwise comparisons (t-test; $p < 0.05$). Lipid species in bold represent features that displayed significant differences in concentration in more than one contrast (G2 x G1 and G3 x G2). Concentrations of selected lipid species are shown in S2 Fig for comparison. HODE* emphasizes that this Oxy-FA was elevated in G3 relative to the other groups. Abbreviations for lipid classes are depicted in Table 1 or in the main text. Fatty acyl chains are represented by X:Y, where X denotes the number of carbons and Y the number of double bonds.

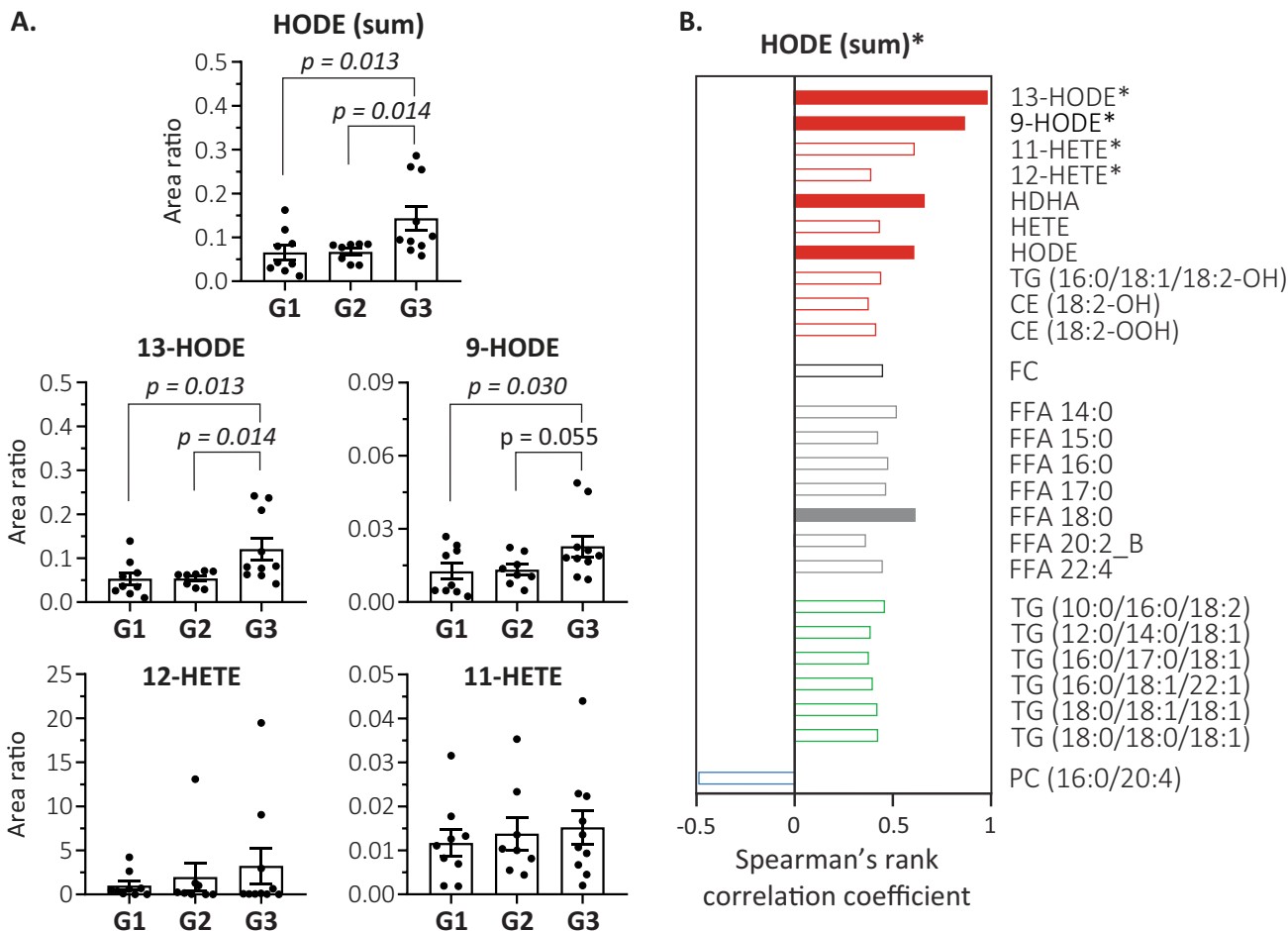

**Fig 3. Relationship between plasma levels of hydroxyoctadecadienoic acid (HODE) and ZIKV-induced microcephaly.** (A) Oxilipidomic analysis of HODE (see also S3 Fig) and hydroxyeicosatetraenoic acid (HETE) isomers (modified from [48]; S2 Methods). HODE (sum) represents summed concentrations of 9- and 13-HODE isomers. Concentrations are displayed as area ratio (bars indicate average ± standard error of the mean) and comparisons were determined by unpaired t-test (p<0.05 shown in italics). (B) Significant association of HODE (sum) with lipid species (p<0.001 filled bars; p<0.05 empty bars) by Spearman's rank correlation analysis. Bars are color coded for oxidized lipids (red), free cholesterol (black), FFA (gray), TG (green) and phospholipid (blue). Oxidized linoleic acid linked to TG and CE is represented by 18:2-OH or -OOH, where OH denotes monohydroxyl and OOH hydroperoxyl groups. * = lipids quantified by oxilipidomics; HDHA = hydroxydocosahexaenoic acid. Abbreviations for lipid classes are depicted in Table 1 or in the main text.

targeted oxilipidomic analysis to assess the contribution of different isomers of HODE and hydroxyeicosatetraenoic acid (HETE) (see details in S2 Methods). The results evidenced higher concentrations of total HODE (defined by the sum of 9- and 13-HODE isomers) in G3 relative to the other groups (Figs 3A and S3), confirming our preliminary assessment. The data further revealed that 13-HODE not only displayed higher concentrations than 9-HODE, but also accounted for the most significant differences between G3 and the other groups. In contrast, HETE isomers concentrations displayed no significant alterations among the groups (Fig 3A). A Spearman's rank correlation analysis (Fig 3B) indicated a positive correlation of total HODE with a number of oxidized lipids (including the Oxy-FA, and oxidized linoleic acid linked to TG and cholesteryl esters), several FFA, free cholesterol and 6 TG, the latter mostly linked to saturated or mono-unsaturated fatty acids. Moreover, negative association of total HODE was evidenced for PC (16:0/20:4).

In summary, our results revealed major alterations in newborns' plasma lipidome linked to ZIKV exposure, particularly modulations involving the linoleic acid. In the ZIKV-infected normocephalic G2, concentrations of several TG esterified to linoleic acid were reduced in comparison to G1, apart from lower free cholesterol and higher LPC. Plasma from newborns with ZIKV-induced microcephaly (G3) was enriched in HODE, particularly the 13-HODE isomer, relative to normocephalic newborns (G1 and G2). In addition, total HODE was significantly correlated with several circulating FFA and oxidized lipids.

## Discussion

Lipids play a central role in metabolism, membrane structure and signaling during early brain development [49]. Their importance may be better exemplified by the cognitive and visual capabilities of infants linked to maternal or cord docosahexaenoic acid (DHA) status in observational studies (reviewed in [50]). The maternal-fetal interface is a major contributor to the quality of lipids delivered to embryos/fetuses. The striking differences in lipid content between the mother's blood plasma and the umbilical cord plasma is a testament of its importance in the transport of lipids across the placenta. For instance, cord blood plasma is enriched by 1.5 to 4 fold in arachidonic acid and DHA content of major lipid classes as compared to mother's plasma [51]. Placental sufficiency is thus vital for a healthy early brain development. While congenital ZIKV syndrome has been attributed to increased death of neural progenitor cells in cellular and mice models [14–16,52–54], the mechanisms of ZIKV infection leading to brain defects in newborns remain unknown. Nonetheless, a staggering number of observational and experimental studies have documented that ZIKV targets the placental cells, resulting not only in increased systemic inflammation [19,23–25], but also significant changes in placental lipid metabolism [29]. The present study is consistent with the latter observations and suggests that antenatal ZIKV exposure leads to significant changes in umbilical cord plasma lipidome of newborns that may reflect neurodevelopmental, neurocognitive and ophthalmological abnormalities beyond microcephaly.

A common link among alterations in plasma lipidome of ZIKV exposed newborns involves the linoleic acid. The role of linoleic acid in pregnancy relies on its concentrations in cord blood plasma that is reduced by 10 fold relative to mother's plasma [51], likely as a result of intense metabolism of this fatty acid during placental development. Although displaying high variability in concentrations of total lipid classes (S1 Fig), our data consistently revealed higher HODE concentrations, particularly the 13-HODE isomer, in plasma from newborns with ZIKV-induced microcephaly relative to the other groups. Oxidized fatty acids, such as HODE, are generated by the activation of oxygenases [55,56] (cyclooxygenases and lipoxygenases mainly, and cytochrome P450 to a minor extent) and by free radical mediated lipid peroxidation [57]. Enzymatic oxidation of fatty acids plays a pivotal role during a normal reproductive cycle and pregnancy [58]. However, elevated plasma HODE concentrations and their positive correlation with other oxidized lipids in ZIKV-induced microcephalic newborns appears more consistently linked to the systemic inflammatory profile and redox imbalanced environment of the ZIKV-infected placenta [22–24,29]. A more direct link of HODE to early neurodevelopment was found for elevated concentrations of 9-HODE in neural stem cells (derived from human embryonic stem cells) infected with cytomegalovirus [59]. These authors have shown that either high concentrations of 9-HODE generated by infected cells or treatment of non-infected cells with 9-HODE led to increased levels and activity of the peroxisome proliferator-activated receptor (PPAR) gamma, which in turn were associated with impaired rates of neurogenesis. Immunodetection of nuclear PPAR-gamma in germinative zones of cytomegalovirus-infected human fetal brain and absence in control fetuses confirmed the role of PPAR-

gamma in congenital neuropathogenesis of cytomegalovirus infection [59]. Similar to congenital ZIKV syndrome, congenital human cytomegalovirus infection is a leading cause of permanent and severe neurological sequelae, including microcephaly and hearing and vision loss [60].

Importantly, plasma concentrations of HODE were positively associated with several FFA, the majority of them found in higher concentrations in newborns with ZIKV-induced microcephaly relative to controls. Moreover, decreased concentrations of TG esterified to linoleic acid were observed in close association with elevated concentrations of lysophosphatidylcholine (LPC) species in normocephalic ZIKV exposed newborns. High circulating FFA and LPC in plasma are in general correlated with elevated hydrolytic activities of lipases, such as the endothelium lipase and lipoprotein lipase found in the placenta [61]. It is known since the 60's that microcephaly is associated with a remarkable accumulation of neutral lipids as cytosolic lipid droplets in glial cells [62]. A combination of high fluxes of FFA and a "leaky" blood-brain barrier due to ZIKV infection [63,64] might contribute to the accumulation of neutral lipids in the central nervous system.

More recently, ZIKV was reported to disrupt the major facilitator superfamily domain-containing protein 2 (Mfsd2a), also known as a membrane bound sodium-dependent LPC symporter, both in human brain endothelial cell cultures and neonatal mouse brain, causing growth restriction and microcephaly in the latter [36]. The importance of this data lies on the pivotal role of Mfsd2a in the transport of DHA across the blood-brain barrier via endothelium cells [33] as well as its role in ensuring integrity of the blood-brain barrier [65]. Two case studies have reported that inactivating mutations in the Mfsd2a (full or partial loss of function) cause human microcephaly and established a correlation between the degree of mutation and the severity of the pathology [34,35]. Notably, both studies also found out that affected individuals displayed high concentrations of circulating plasma LPC, especially those esterified to mono- and polyunsaturated fatty acids, suggesting a major role of Mfsd2a in the transport of these fatty acids to the brain. The increased plasma concentrations of LPC in the normocephalic ZIKV exposed newborns compared to controls undoubtedly did not lead to apparent alterations in head circumference at birth. However, recent data from follow up studies have shown a high frequency of neurodevelopmental and ophthalmological abnormalities in children with antenatal exposure to ZIKV and normocephalic at birth [2,10–13]. The clinical outcomes are remarkable, for instance, 68% of those children affected by neurological abnormalities on physical examination, 30% with abnormal neuroimaging, and 57% with complications to thrive given their poor feeding neurological capabilities [13]. Our study suggests that plasma lipidome profiling in newborns exposed to ZIKV may potentially contribute to recognize clinical phenotypes linked to abnormal neurodevelopment observed at birth and during the course of their childhood.

In summary, the observations reported in our study are in line with evidence that ZIKV infection disturbs the homeostasis in placental cells, which are responsible for the selective transport of lipids from the mother's blood to the cord blood of their fetuses. This transport is dictated by the stage of development of the fetuses, thus the requirements of specific lipids are distinct in each trimester of gestation [66]. Placental insufficiency is a major cause of fetal growth restriction with severe consequences to early neurodevelopment [67]. We suspect that our data reflect the diverse mechanisms by which antenatal ZIKV exposure negatively impact the neurodevelopment of fetuses/infants, especially the spatial-temporal patterns based not only on the trimester of pregnancy, but also on the intensity of infection in each specific cells/tissues.

Our study has several limitations. Firstly, samples were taken at a single point, thus changes occurring before birth are missing, and lack information regarding mother's fasting or post-prandial conditions. Moreover, major redox and inflammatory changes occurring in the

placenta during labor [68] are expected to lead to significant fluctuations in lipid concentrations in cord fluids. The great extent of variation in the concentrations of lipid classes and distribution of major fatty acids (S1 Fig) might have stemmed from the latter points. For instance, there was a trend of lower cholesteryl esters, together with significant changes in free cholesterol, in the normocephalic ZIKV exposed group (S1A Fig). Cholesteryl esters are the most abundant lipid class in newborn's plasma [46] and alterations in this pool may result in a considerable decrease in fatty acids being transported to the infants. Secondly, quantification of HODE isomers revealed that concentrations of 13-HODE, and to a lesser extent 9-HODE, are elevated in plasma from newborns with ZIKV-induced microcephaly. While a direct role of 9-HODE impairing neurogenesis in stem cells infected by cytomegalovirus has been established [59], a causal link between HODE isomers and the neuropathogenesis of ZIKV remains to be investigated. Thirdly, although controlling for potential confounders, the number of individuals selected for this study was relatively small and thus a second or larger cohort is required to validate our results. Nonetheless, our study identified significant changes in cord plasma lipidome associated with antenatal ZIKV exposure that may contribute to detection of the wide spectrum of clinical phenotypes observed at birth and later in childhood. Further prospective studies with a larger cohort of newborns are now required for validation of the predictive ability of this approach.

## Supporting information

**S1 Fig. Quantification of major lipid classes and major polyunsaturated fatty acids composition of total lipids and lipid classes.** (A) Total lipid classes concentrations in mg dL$^{-1}$ in G1 (non-infected controls), G2 (normocephalic ZIKV-infected) and G3 (ZIKV exposed microcephalic). (B) Total concentrations of polyunsaturated fatty acids in mM, focusing on linoleic (LA, 18:2), arachidonic (ARA; 20:4) and docosahexaenoic (DHA; 22:6) acids; (C) Distribution of LA, ARA and DHA into major lipid classes. The data shows average ± standard error. Statistical analysis by t-test revealed no alteration regarding quantity or contribution of these lipids among groups, although several trends could be observed, such as lower quantities of CE and total 18:2 in G2 relative to the other groups. For abbreviations of lipid classes please refer to Table 1 and GPL = glycerophospholipids.
(EPS)

**S2 Fig. Concentrations of selected lipid species displaying significant alterations among groups (as shown in Fig 1).** Bars represent the average concentration of lipids (ng/μL) ± standard error of the mean. For significance, unpaired t-test was applied with brackets indicating p<0.05.
(EPS)

**S3 Fig. Structure confirmation of HODE isomers by targeted oxilipidomics.** (A) Extracted ion chromatograms of HODE isomers as standards and in a representative experimental sample. MS/MS spectra of 13-HODE (B) and 9-HODE (C) standards used to define specific fragment ions for quantification of HODE isomers.
(EPS)

**S1 Table. Epidemiological description of samples.**
(DOCX)

**S2 Table. Integral lipidomics data in area ratio concentrations, including experimental and quality control samples.**
(CSV)

**S3 Table. Identification of lipids by MS/MS experiments: Lipid identification, major ions in both positive and negative ionization mode, retention times and exact masses.**
(XLSX)

**S1 Methods. Detailed description of methods applied to plasma lipidome profiling.**
(DOCX)

**S2 Methods. Targeted oxilipidomic analysis of HODE and HETE.**
(DOCX)

## Acknowledgments

The authors thank all personnel from the public maternity hospital Prof. José Maria Magalhães Netto (State Health Secretary, Salvador, Bahia, Brazil) who coordinated surveillance and helped with data collection and assembly. Cleiton Santos from the Gonçalo Moniz Institute (FIOCRUZ, Rio de Janeiro, Brazil) is acknowledged for laboratory technical assistance as well as Dr. Ester Sabino and collaborators at the Tropical Medicine Institute of the University of São Paulo (São Paulo, Brazil) for kindly providing laboratory space for sample preparation.

## Author Contributions

**Conceptualization:** Nieli Rodrigues da Costa Faria, Adriano Britto Chaves-Filho, Isadora Cristina de Siqueira, Marcos Yukio Yoshinaga.

**Data curation:** Nieli Rodrigues da Costa Faria, Luiz Carlos Junior Alcantara, Isadora Cristina de Siqueira, Juan Ignacio Calcagno.

**Formal analysis:** Nieli Rodrigues da Costa Faria, Adriano Britto Chaves-Filho, Marcos Yukio Yoshinaga.

**Funding acquisition:** Sayuri Miyamoto, Ana Maria Bispo de Filippis.

**Investigation:** Nieli Rodrigues da Costa Faria, Adriano Britto Chaves-Filho, Luiz Carlos Junior Alcantara, Isadora Cristina de Siqueira, Juan Ignacio Calcagno, Ana Maria Bispo de Filippis, Marcos Yukio Yoshinaga.

**Methodology:** Adriano Britto Chaves-Filho, Sayuri Miyamoto, Marcos Yukio Yoshinaga.

**Resources:** Luiz Carlos Junior Alcantara, Isadora Cristina de Siqueira, Juan Ignacio Calcagno, Sayuri Miyamoto, Ana Maria Bispo de Filippis.

**Supervision:** Ana Maria Bispo de Filippis, Marcos Yukio Yoshinaga.

**Writing – original draft:** Nieli Rodrigues da Costa Faria, Adriano Britto Chaves-Filho, Marcos Yukio Yoshinaga.

**Writing – review & editing:** Luiz Carlos Junior Alcantara, Isadora Cristina de Siqueira, Juan Ignacio Calcagno, Sayuri Miyamoto, Ana Maria Bispo de Filippis.

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
