## [Decision Letter · Decision Letter 0]

18 Mar 2021

Dear Dr. Yoshinaga,

Thank you very much for submitting your manuscript "Plasma lipidome profiling of newborns with antenatal exposure to Zika virus" for consideration at PLOS Neglected Tropical Diseases. As with all papers reviewed by the journal, your manuscript was reviewed by members of the editorial board and by several independent reviewers. In light of the reviews (below this email), we would like to invite the resubmission of a significantly-revised version that takes into account the reviewers' comments. 

The reviewers found this to be a well written and executed study, but did have some suggestions that could strengthen the manuscript. One reviewer asked for an additional experiment that seems reasonable and should be highly considered by the authors.

We cannot make any decision about publication until we have seen the revised manuscript and your response to the reviewers' comments. Your revised manuscript is also likely to be sent to reviewers for further evaluation.

Sincerely,

Doug E Brackney, PhD

Associate Editor

Rebecca Rico-Hesse

Deputy Editor

The reviewers found this to be a well written and executed study, but did have some suggestions that could strengthen the manuscript. One reviewer asked for an additional experiment that seems reasonable and should be highly considered by the authors.

Reviewer's Responses to Questions

**Key Review Criteria Required for Acceptance?**

**Methods**

-Are the objectives of the study clearly articulated with a clear testable hypothesis stated?

-Is the study design appropriate to address the stated objectives?

-Is the population clearly described and appropriate for the hypothesis being tested?

-Is the sample size sufficient to ensure adequate power to address the hypothesis being tested?

-Were correct statistical analysis used to support conclusions?

-Are there concerns about ethical or regulatory requirements being met?

Reviewer #1: The authors investigate lipidomic profile of ZIKA congenitally infected nowborns by using high performance liquid chromatography. The method suffers from two limitations (unability to diciminate HODE isoforms and single point sampling) but they are discussed and do not critically change the manuscript message.

Reviewer #2: (No Response)

Reviewer #3: This study compares the plasma lipidome profiles of Zika virus exposed microcephalic and normocephalic newborns compared to uninfected controls. A key oxidized lipid, hydroxyoctadeconoic acid (HODE) derived from linoleic acid was observed to be higher in microcephalic newborns compared to the norpmocephalic group and controls. HODE was also positively associated with increased circulating free fatty acids providing a preliminary basis for a plasma lipidome signature. An additional interesting finding was the observation that lysophosphatidylcholine was higher in ZIKV exposed normocephalic newborns linking possibilities that polyunsaturated fatty acid transport may be hindered. This study is very well written and results clearly explained. The lipidomics and the statistical analyses are clearly carried out and written well in the materials and methods. Concerns regarding sample pool size is addressed in the discussion including other limitations. No concerns about ethical or regulatory requirements.

**Results**

-Does the analysis presented match the analysis plan?

-Are the results clearly and completely presented?

-Are the figures (Tables, Images) of sufficient quality for clarity?

Reviewer #1: The data sound and are presented clearly. Figures and tables are well designed and allow nice understanding of the results.

Reviewer #2: (No Response)

Reviewer #3: The results are clearly explained and limitations addressed. 

It would be very valuable to the readership if the authors could also provide a table of information of all the data from the untargeted analyses with included observed mass, retention time, putative IDs, accurate mass, ppm error etc so that a complete picture of the observed plasma lipidome of these newborns could be assessed with the understanding that accurate identification of the masses require further analyses.

**Conclusions**

-Are the conclusions supported by the data presented?

-Are the limitations of analysis clearly described?

-Do the authors discuss how these data can be helpful to advance our understanding of the topic under study?

-Is public health relevance addressed?

Reviewer #1: (No Response)

Reviewer #2: (No Response)

Reviewer #3: The conclusions based on this limited dataset are valid and are well discussed compared to the current literature.

**Editorial and Data Presentation Modifications?**

Reviewer #1: There is a reference displayed as "Kikut et al., 2020", line 281, unlike the others, displayed as numbers.

Reviewer #2: (No Response)

Reviewer #3: (No Response)

**Summary and General Comments**

Reviewer #1: Viral infection during pregnacy may associate with alteration of lipid metabolism, thereby threatening homeostasis and development. Here the authors show that the lipidomic profile of newborns congenitally infected by ZIKV differs from that of control subjects and show increased HODE levels. The findings provide new insight on ZIKV infection pathophysiology. They disclose new clues to understand neurological sequelae since elevated HODE levels have been already associated with HCMV infection, even though no causality can be formally established.

Reviewer #2: (No Response)

Reviewer #3: While this is a limited data set and limited number of samples that are challenging to acquire, the information resulting from this study are clearly presented and can drive future analyses of how lipids might control newborn brain development as well as how ZIKV might alter this process.

PLOS authors have the option to publish the peer review history of their article (what does this mean?). If published, this will include your full peer review and any attached files.

Reviewer #1: No

Reviewer #2: No

Reviewer #3: No
---

## [Editor Report · Decision Letter 1]

14 Apr 2021

Dear Dr. Yoshinaga,

I have reviewed the revisions and added material and don't believe it requires being disseminated to reviewers for a second round of reviews. Therefore, we are pleased to inform you that your manuscript 'Plasma lipidome profiling of newborns with antenatal exposure to Zika virus' has been provisionally accepted for publication in PLOS Neglected Tropical Diseases.

Best regards,

Doug E Brackney, PhD

Associate Editor

Rebecca Rico-Hesse

Deputy Editor

---

## [Editor Report · Acceptance letter]

27 Apr 2021

Dear Dr. Yoshinaga,

We are delighted to inform you that your manuscript, "Plasma lipidome profiling of newborns with antenatal exposure to Zika virus," has been formally accepted for publication in PLOS Neglected Tropical Diseases.

Best regards,

Shaden Kamhawi

co-Editor-in-Chief

Paul Brindley

co-Editor-in-Chief
